# Trends of proximate low birth weight and associations among children under-five years of age: Evidence from the 2016 Ethiopian demographic and health survey data

Mesfin Wudu Kassaw[1]*, Ayele Mamo Abebe[2], Ayelign Mengesha Kassie[1], Biruk Beletew Abate[1], Seteamlak Adane Masresha[3]

1 Department of Nursing, College of Health Science, Woldia University, Woldia, Ethiopia, 2 Department of Nursing, College of Health Science, Debre Berhan University, Debre Berhan, Ethiopia, 3 Department of Public Health, College of Health Science, Woldia University, Woldia, Ethiopia

* mesfine12a@gmail.com

## Abstract

### Background

Low birth weight puts a newborn at increased risk of death and illness, and limits their productivity in the adulthood period later. The incidence of low birth weight has been selected as an important indicator for monitoring major health goals by the World Summit for Children. The 2014 World Health Organization estimation of child death indicated that 4.53% of total deaths in Ethiopia were due to low birth weight. The aim of this study was to assess trends of proximate low birth weight and associations of low birth weight with potential determinants from 2011 to 2016.

### Methods

This study used the 2016 Ethiopian Demographic and Health Survey data (EDHS) as data sources. According to the 2016 EDHS data, all the regions were stratified into urban and rural areas. The variable "size of child" measured according to the report of mothers before two weeks of the EDHS takes placed. The study sample refined from EDHS data and used for this further analysis were 7919 children. A logistic regression model was used to assess the association of proximate low birth weight and potential determinates of proximate low birth weight. But, the data were tested to model fitness and were fitted to Hosmer-Lemeshow-goodness of fit.

### Results

The prevalence of proximate low birth weight in Ethiopia was 26.9% (2132), (95%CI = 25.4, 27.9). Of the prevalence of child size in year from 2011 to 2016, 17.1% was very small, and 9.8% was small. In the final multivariate logistic regression model, region (AOR = xx), (955%CI = xx), Afar (AOR = 2.44), (95%CI = 1.82, 3.27), Somalia (AOR = 0.73), (95%CI = 0.55, 0.97), Benishangul-Gumz (AOR = 0.48), (95%CI = 0.35, 0.67), SNNPR (AOR = 0.67),

**Data Availability Statement:** All relevant data are within the paper.

**Funding:** The authors received no specific funding for this work.

**Competing interests:** The authors have declared that no competing interests exist.

**Abbreviations:** AOR, adjusted odds ratio; CI, confidence interval; SPSS, statistical package for social science; EDHS, Ethiopian Demographic and Health Survey data; EA, Enumeration areas; SNNPR, Southern nations, nationalities, and people representative of Ethiopia; ICF, inner city fund; LBW, Low birth weight; WHO, World Health Organization; KMC, Kangaroo mother care.

(95%CI = 0.48, 0.93), religion, Protestant (AOR = 0.76), (95%CI = 0.60, 0.95), residence, rural (AOR = 1.39), (95%CI = 1.07, 1.81), child sex, female (AOR = 1.43), (95%CI = 1.29, 1.59), birth type, multiple birth during first parity (AOR = 2.18), (95%CI = 1.41, 3.37), multiple birth during second parity (AOR = 2.92), (95%CI = 1.86, 4.58), preparedness for birth, wanted latter child (AOR = 1.26), (95%CI = 1.09, 1.47), fast and rapid breathing (AOR = 1.22), (95%CI = 1.02, 1.45), maternal education, unable to read and write (AOR = 1.46), (95%CI = 1.56, 2.17), and maternal age, 15–19 years old (AOR = 1.86), (95%CI = 1.19, 2.92) associated with proximate low birth weight.

## Conclusions

The proximate LBW prevalence as indicated by small child size is high. Region, religion, residence, birth type, preparedness for birth, fast and rapid breathing, maternal education, and maternal age were associated with proximate low birth weight. Health institutions should mitigating measures on low birth weight with a special emphasis on factors identified in this study.

## Introduction

World Health Organization (WHO) defines low birth weight (LBW), when a weight at birth is less than 2500gm [1]. LBW puts a newborn at increased risk of death and illness [2–4], and limits their productivity in the adulthood period [5]. Worldwide, 15 to 20% of newborns were low-birth weight (LBW) [6]. The prevalence of LBW in sub-Saharan Africa was 13% [6]. The incidence of LBW has been selected as important indicator for monitoring major health goals by the World Summit for Children [7]. In most developed countries, data on birth weight are available for deliveries that took place in health facilities. However, in developing countries, data on birth weight are difficult to obtain, as most births occur outside health facilities. Many infants are not weighed at the time of birth [8]. In such low income countries, women giving birth in a hospital are likely to have a higher socio-economic status than women who deliver at home and might be less likely to give birth to LBW babies [9–11]. In between 2000 and 2015, there was no region that can decrease low birth weight prevalence globally to a significant level. Evidence has shown that the prevalence of LBW had been reduced by only 1.2% each year in the year 2000 to 2015 worldwide [12], which is suggesting an insufficient progress towards the 2025 World Health Assembly low birth weight target [13]. The burden of LBW in Ethiopia is nearly half of the Sub-Saharan Africa (SSA) average, 11% [14]. Another study also reported that the prevalence of LBW in Ethiopia was 10.4% [15]. This might be why Ethiopia was the one among five countries those account for about half of all global neonatal deaths [16]. The complication of LBW is profound in Ethiopia, even death. For example, in 2017 Low birth weight accounted for 3.63% of total deaths according to WHO report [17]. Even though, data on low birth weight was still limited to assess trends and to compare annual variations due to low institutional delivery in Ethiopia, the prevalence of low birth weight was increased by 5% from 2000 [18], to 2016 [19]. Furthermore, the evidence from systematic review and meta-analysis showed that the pooled prevalence of low birth weight in Ethiopia was 17.3% [20]. Despite such high prevalence of LBW, the 2011 Ethiopian Demographic and Health Survey data indicated that only 5% of children were weighed at birth [14,21]. Thus, birth weight data from health facilities has limited usage in assessing the prevalence, and analyzing determinants of LBW in developing countries [9,10]. But LBW predispose newborns to may

complications like hypoglycemia [22], hypothermia [23], mental retardation, physical, and neurodevelopmental problems [24], as well as the risk of death is high in a LBW infants. The 2014 World Health Organization (WHO) estimation indicated that 4.53% of total deaths in Ethiopia [20] were due to LBW. The 2011 demographic health survey of Ethiopia (EDHS) showed that 29% of Ethiopian babies weigh low as perceived by their mothers [14]. A study from America reported that LBW babies were twenty times more likely to neonatal morbidity than babies weighing >2500gm [25,26]. Evidences also revealed that LBW babies are likely to have disabilities such as poor schooling, hospitalization, poor language development and intellectual impairments. A number of studies showed that the risk factors for LBW relate to the fetus, maternal health, and the environment in which the mother lives [27–29]. Although the current study considers mothers' perception of their children size at birth as outcome variable, there was evidence that mothers' perception of their child weight were proxy to a measured weight. The evidence recommends that any estimates of LBW on the basis of maternal recalled birth size should be considered as underestimates of the actual prevalence [30]. The above study showed that 92% of mothers provided concordant results on the child size (small and normal) they recalled was in agreement with the category of recorded birth weight (LBW and normal) [30]. Despite this high low birth weight, the Ethiopian government aspired to decrease the prevalence of LBW using a number of implementations. These stated implementations include KMC, early initiation of breastfeeding, post natal care for the mother, post natal care for the newborn, exclusive breastfeeding and etc. [31]. The aim of this study was to assess the trends of proximate LBW and associations from 2011 to 2016 in Ethiopia using the 2016 EDHS data. Since the survey had 5 years compressive data, it can indicate the trends better than the other primary and area specific studies. Because of that most previous conducted studies are limited geographically, by population, and sample size. In addition, those studies are single shot and did not show the trends of low birth weight which might be susceptible to bias because of drought or other seasonal factors like political instability and natural disaster. However, this study considered a 5 consecutive year's data to assess the trends and potential determinants of proximate low birth weight.

## Methods

### Data collection period, study design, and data collection

The data collection period for the 2016 EDHS was from January 18 to June 27, 2016. The 2016 Ethiopian Demographic and Health Survey (EDHS) data was used in this further analysis. The 2016 EDHS data was the fourth survey conducted in Ethiopia. The survey collected information on household's and respondent's characteristics, child health, infant and child mortality, malaria, maternal health, maternal mortality, nutrition, tobacco use, women's empowerment, anemia, domestic violence, environmental health, family planning, female genital cutting, fertility and fertility preferences, and etc.

The purpose of the EDHS is to provide up-to-date estimates of the key demographic and health indicators of the population [32]. The survey included reproductive age group women, under-five children, and productive age group men (aged 15–59 years) [32,33].

### Sampling technique and study population

The 2016 EDHS data collected using a stratified two stage sampling method to select a representative sample. All the regions of the country were stratified into urban and rural areas. From total 11 administrative states, 21 sampling strata were yielded. The samples of enumeration areas (EAs) were selected independently in each stratum in two stages. The implicit stratification and proportional allocation were achieved at each of the lower administrative levels by

sorting the sampling frame within each sampling stratum before sample selection according to the administrative units in a different level, and by using a probability proportion to size selection at the first stage of sampling. The 2016 EDHS selected 645 EAs with a probability proportional to the EA size and with independent selection in each sampling stratum. The EA size is the number of residential households in the EA that was determined in the 2007 Ethiopian Population and Housing Census. According to the 2016 EDHS procedures, a household listing operation was implemented in the selected EAs, and the resulting lists of households served as the sampling frame for the selection of households in the second stage. The data collectors interviewed only the pre-selected households. In the EDHS, there were no replacements or changes of the pre-selected households in the implementing stages to prevent bias. All the under-five children, who were usual members of the selected households or who spent the night before the survey in the selected households were eligible for the child survey [33]. After managing the missing data, 7919 respondents that had under-five children were included for this further analysis. However, the 2016 EDHS data collected from 10641 children. Yet, the original EDHS data were undergoing through rigorous phases of data refining until we get the final sample size, 7919.

## Data collection tools and procedures

The EDHS usually use five groups of questionnaires in collecting the data. Those questionnaires are the Household questionnaire, the woman's questionnaire, the man's questionnaire, the biomarker questionnaire, and the health facility questionnaire. The questionnaires were adapted from the DHS program's standard demographic and health survey questionnaires in a way to reflect the population and health issues relevant to Ethiopia. Questions that stated about children were integrated to woman's questionnaire.

## Ethics approval and consent to participate

The ethical clearance was obtained from the Institutional Review Board of Woldia University. A support letter was also obtained from Woldia University, research directorate office. Written informed consent was obtained by EDHS data collectors from mothers who were involved in the study after explaining the aim of the study. Anonymity and confidentiality were maintained by allowing opposition and or discontinuation of the interview and omitting the name and personal identification of respondents, because it was not relevant to the survey. In addition, the study was followed guidelines outlined in the Declaration of Helsinki.

## Study variables

The outcome variable of this further analysis was proximate low birth weight. The independent variables were socio-demographic variables of both children and mothers, health services provided to children and the wider community, and substance use like Cigarette smoking and Khat chewing in considering the availability of those variables in the 2016 EDHS database.

## Measurement of study variables

The variable "size of child" measured according to the report of mothers before two weeks of the EDHS takes placed. Mothers reported that their child was classified as very large, large, average, small, and very small. This classification of child size were re-coded to average and above (very large, large, and normal), and below average (small, and very small). This below average classification in this study considered as proximate low birth weight.

### Definition

**Child size.** The size of the child at birth according to the mothers' personal evaluation in relative to other children from their experience.

### Data analysis

The data analysis started with a summary of socio-demographic characteristics using both descriptive and inferential statistics. The trends of proximate low birth weight presented in percent for the years 2011 to 2016 to assess the downgrade or upgrade of LBW. A logistic regression model was used to assess the association of proximate low birth weight and independent variables. The data were assessed against the assumption of logistic regression like normality, sample size and model fitness. In this regard, the data were fitted to Hosmer-Lemeshow-goodness of fit, and normality test. In the binary logistic regression, variables that got a P-value of 0.25 and below were selected to multivariate logistic regression analysis. The equation used for logistic model was $\log\left(\frac{\pi}{1-\pi}\right) = \beta_0 + \beta_1 X$. A statistically significant association was determined at a p-value of less than 0.05.

## Results

### Socio-demographic status

In this study, 7919 mothers who have under-five children were included from the 2016 EDHS data after refining. Of the include mothers, 30.2% (2395) of mothers were in the age group of 25–29. In the study, 83.7% (6628) of mothers and their children were from rural areas, 57.6% (4558) of the participants were had agricultural job, and 50.9% (4027) of mothers were a follower of Muslim religion. Regarding health care service 74.9% (5934) of women did not practice KMC, and 90.7% (7180) of women reported that their children had no history of fast and rapid breathing. In addition, 65.6% (5195) of mothers have no formal educational status, and 79.8% (6322) of women wanted their last birth (Table 1).

### Trends and prevalence of proximate low birth weight

The study indicates that 7919 children were born from 2011 to 2016. In particular, 6.3% (500) children were born in 2011, 20.2% (1603) in 2012, 19.4% (1537) in 2013, 19.3% (1532) in 2014, 21.6% (1707) in 2015, and 13.1% (1040) in 2016. The prevalence of very small and small child size were 21.5% (224) and 11.3% (117) in 2016, 18% (307) and 9.1% (156) in 2015, 17% (260) and 9.8% (150) in 2014, 16.1% (248) and 10% (154) in 2013, 15.8% (253) and 9.5% (153) in 2012, and 12.8% (64) and 9.2% (46) in 2011 respectively (Fig 1). The prevalence of proximate low birth weight in Ethiopia was 26.9% (2132). The proximate low birth weight by year from 2011 to 2016 indicated that 17.1% (1356) was very small, and 9.8% (776) was small (Fig 2).

### Associations with proximate low birth weight

Variables that had a P-value of 0.25 and below were included in the multivariate regression. But, on the binary logistic regression model, only wealth index, region, residence, fast and rapid breathing, place of delivery, KMC practice, maternal age, religion, sex of head of household, child sex, child birth type, and maternal education associated with proximate LBW. However, in the multivariate logistic regression model, region (AOR = xx), (955%CI = xx), Afar (AOR = 2.44), (95%CI = 1.82, 3.27), Somalia (AOR = 0.73), (95%CI = 0.55, 0.97), Benishangul-Gumz (AOR = 0.48), (95%CI = 0.35, 0.67), SNNPR (AOR = 0.67), (95%CI = 0.48, 0.93), religion, Protestant (AOR = 0.76), (95%CI = 0.60, 0.95), residence rural (AOR = 1.39),

**Table 1. The socio-demographic, behavioral, and medical history of respondents and their children under-five years of age in Ethiopia.**

| Variables | Categories | Frequency (n = 7919) | Percent |
|---|---|---|---|
| Husband occupation | Did not work | 870 | 11.0 |
| | Professional/managerial | 598 | 7.6 |
| | Clerical | 56 | 0.7 |
| | Sales | 650 | 8.2 |
| | Agricultural employee | 4558 | 57.6 |
| | Service | 304 | 3.8 |
| | Skilled manual | 533 | 6.7 |
| | Unskilled manual | 350 | 4.4 |
| Maternal occupation | Did not work | 4786 | 60.4 |
| | Professional/managerial | 126 | 1.6 |
| | Clerical | 35 | 0.4 |
| | Sales | 819 | 10.3 |
| | Agricultural employee | 1590 | 20.1 |
| | Service | 138 | 1.7 |
| | Skilled manual | 237 | 3.0 |
| | Unskilled manual | 105 | 1.3 |
| Head of household | Male | 6602 | 83.4 |
| | Female | 1317 | 16.6 |
| Residence | Urban | 1291 | 16.3 |
| | Rural | 6628 | 83.7 |
| Maternal age | 15–19 | 265 | 3.3 |
| | 20–24 | 1573 | 19.9 |
| | 25–29 | 2395 | 30.2 |
| | 30–34 | 1812 | 22.9 |
| | 35–39 | 1237 | 15.6 |
| | 40–44 | 483 | 6.1 |
| | 45–49 | 154 | 1.9 |
| Maternal education | No education | 5195 | 65.6 |
| | Primary | 1978 | 25.0 |
| | Secondary | 492 | 6.2 |
| | Higher | 254 | 3.2 |
| Region | Tigray | 757 | 9.6 |
| | Afar | 781 | 9.9 |
| | Amhara | 805 | 10.2 |
| | Oromia | 1255 | 15.8 |
| | Somali | 1071 | 13.5 |
| | Benishangul | 663 | 8.4 |
| | SNNPR | 1036 | 13.1 |
| | Gambella | 463 | 5.8 |
| | Harari | 422 | 5.3 |
| | Addis-Ababa | 301 | 3.8 |
| | Dire-diwa | 365 | 4.6 |

(*Continued*)

**Table 1.** (Continued)

| Variables | Categories | Frequency (n = 7919) | Percent |
|---|---|---|---|
| Religion | Orthodox | 2312 | 29.2 |
| | Catholic | 49 | 0.6 |
| | Protestant | 1400 | 17.7 |
| | Muslim | 4027 | 50.9 |
| | Traditional | 70 | 0.9 |
| | Other | 61 | 0.8 |
| Wealth index | Poorest | 2903 | 36.7 |
| | Poorer | 1426 | 18.0 |
| | Middle | 1142 | 14.4 |
| | Richer | 1010 | 12.8 |
| | Richest | 1438 | 18.2 |
| Readiness | Wanted then | 6322 | 79.8 |
| | Wanted later | 1129 | 14.3 |
| | Wanted no more | 468 | 5.9 |
| Cigarette smoking | No | 7843 | 99.0 |
| | Yes | 76 | 1.0 |
| Chat chewing | No | 7080 | 89.4 |
| | Yes | 839 | 10.6 |
| Type of birth | Single birth | 7741 | 97.8 |
| | $1^{st}$ of multiple | 93 | 1.2 |
| | $2^{nd}$ of multiple | 85 | 1.1 |
| Sex of the child | Male | 4058 | 51.2 |
| | Female | 3861 | 48.8 |
| Marital status | Married | 7849 | 99.1 |
| | Living with partner | 70 | 0.9 |
| Fast, rapid breathing | No | 7180 | 90.7 |
| | Yes | 739 | 9.3 |
| Size of the child at birth | Very large | 1255 | 15.8 |
| | Large | 1136 | 14.3 |
| | Average | 3396 | 42.9 |
| | Small | 776 | 9.8 |
| | Very small | 1356 | 17.1 |
| KMC (Kangaroo mother care) | No | 5934 | 74.9 |
| | Yes | 1985 | 25.1 |

(95%CI = 1.07, 1.81), child sex, female (AOR = 1.43), (95%CI = 1.29, 1.59), multiple birth during first parity (AOR = 2.18), (95%CI = 1.41, 3.37), multiple birth during second parity (AOR = 2.92), (95%CI = 1.86, 4.58), preparedness for birth, wanted latter (AOR = 1.26), (95% CI = 1.09, 1.47), fast and rapid breathing, (AOR = 1.22), (95%CI = 1.02, 1.45), maternal education, unable to read and write (AOR = 1.46), (95%CI = 1.56, 2.17), and maternal age, 15–19 years old (AOR = 1.86), (95%CI = 1.19, 2.92) associated with proximate low birth weight (Table 2).

## Discussion

The aim of this study was to assess the trends of proximate low birth weight and association among under-five children in Ethiopia. This study's data is subjective as we used the mothers'

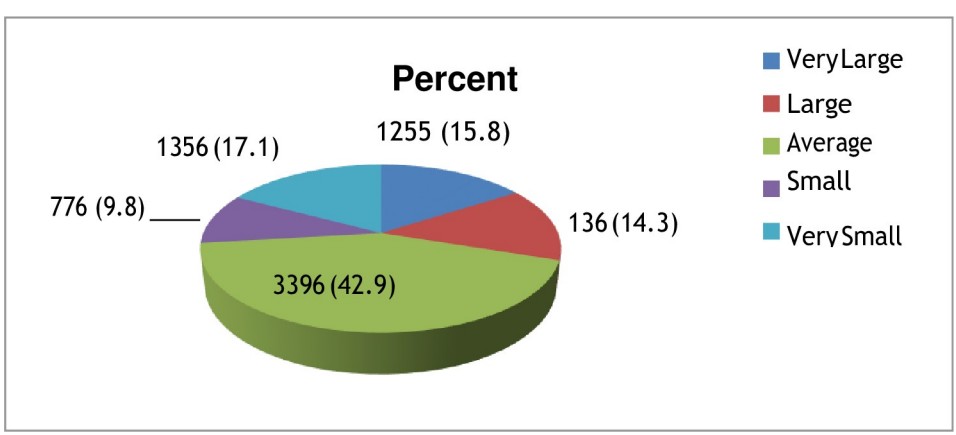

Note: In the Y-axis, the year indicates the Gregorian calendar by Ethiopian Calendar.

**Fig 1. The trends of proximate low birth weight from 2011 to 2016 in Ethiopia.**

**Fig 2. The prevalence of proximate low birth weight in Ethiopia as reported from mothers.**

**Table 2. The association of proximate low birth weight and potential determinants in Ethiopia.**

| Variable | Categories | Proximate low birth weight(n = 7919) | | OR | | 95% CI | | P-value |
|---|---|---|---|---|---|---|---|---|
| | | Average and above | Small to average | COR | AOR | Lower | Upper | |
| Wealth index | Poorest | 1927 (66.4) | 976 (33.6) | 2.25* | 1.25 | 0.96 | 1.62 | 0.10 |
| | Poorer | 1052 (73.8) | 374 (26.2) | 1.58* | 1.18 | 0.90 | 1.54 | 0.23 |
| | Medium | 844 (73.9) | 298 (26.1) | 1.57* | 1.20 | 0.92 | 1.58 | 0.18 |
| | Richer | 790 (78.2) | 220 (21.8) | 1.24* | 0.10 | 0.76 | 1.31 | 0.98 |
| | Richest | 1174 (81.6) | 264 (18.4) | 1 | 1 | 1 | 1 | 1 |
| Region 2020986 | Tigray | 574 (75.8) | 183 (24.2) | 0.83* | 0.78 | 0.55 | 1.09 | 0.14 |
| | Afar | 357 (45.7) | 424 (54.3) | 3.10* | 2.44 | 1.82 | 3.27 | 0.001 |
| | Amhara | 536 (66.6) | 269 (33.4) | 1.31* | 1.09 | 0.79 | 1.52 | 0.59 |
| | Oromia | 934 (74.4) | 321 (25.6) | 0.90 | 0.75 | 0.57 | 1.00 | 0.05 |
| | Somalia | 807 (75.4) | 264 (24.6) | 0.86 | 0.73 | 0.55 | 0.97 | 0.03 |
| | Benishangul | 552 (83.3) | 111 (16.7) | 0.53* | 0.48 | 0.35 | 0.67 | 0.001 |
| | SNNPR | 830 (80.1) | 206 (19.9) | 0.65* | 0.67 | 0.48 | 0.93 | 0.02 |
| | Gambella | 363 (78.4) | 100 (21.6) | 0.72* | 0.80 | 0.55 | 1.16 | 0.24 |
| | Harari | 322 (76.3) | 100 (23.7) | 0.81* | 0.83 | 0.60 | 1.16 | 0.29 |
| | Addis-Ababa | 248 (82.4) | 53 (17.6) | 0.56* | 0.94 | 0.62 | 1.44 | 0.78 |
| | Dire-diwa | 264 (72.3) | 101 (27.7) | 1 | 1 | 1 | 1 | 1 |
| Residence | Rural | 4725 (71.3) | 1903 (28.7) | 1.87* | 1.39 | 1.07 | 1.81 | 0.01 |
| | Urban | 1062 (82.3) | 229 (17.7) | 1 | 1 | 1 | 1 | 1 |
| Smoking | No | 5726 (73.0) | 2117 (27.0) | 1 | 0.62 | 0.34 | 1.13 | 0.12 |
| | Yes | 61 (80.3) | 15 (19.7) | 0.67* | 1 | 1 | 1 | 1 |
| Religion | Orthodox | 1717 (74.3) | 595 (25.7) | 1 | 1 | 1 | 1 | 1 |
| | Catholic | 43 (87.8) | 6 (12.2) | 0.40* | 0.42 | 0.17 | 1.03 | 0.06 |
| | Protestant | 1136 (81.1) | 264 (18.9) | 0.67* | 0.76 | 0.60 | 0.95 | 0.02 |
| | Muslim | 2795 (69.4) | 1232 (30.6) | 1.27* | 0.95 | 0.78 | 1.16 | 0.63 |
| | Traditional | 52 (74.3) | 18 (25.7 | 0.10 | 1.03 | 0.58 | 1.84 | 0.92 |
| | Other | 44 (72.1) | 17 (27.9) | 1.12 | 1.24 | 0.68 | 2.26 | 0.48 |
| Sex of Head of household | Male | 4865 (73.7) | 1737 (26.3) | 1 | 1 | 1 | 1 | 1 |
| | Female | 922 (70.0) | 395 (30.0) | 1.20* | 1.10 | 0.95 | 1.27 | 0.21 |
| Was child wanted | Wanted then | 4644 (73.5) | 1678 (26.5) | 1 | 1 | 1 | 1 | 1 |
| | Wanted latter | 801 (70.9) | 328 (29.1) | 1.13* | 1.26 | 1.09 | 1.47 | 0.002 |
| | Wanted no more | 342 (73.1) | 126 (26.9) | 1.02 | 1.19 | 0.95 | 1.50 | 0.13 |
| Khat chewing | No | 5190 (73.3) | 1890 (26.7) | 1 | 1 | 1 | 1 | 1 |
| | Yes | 597 (71.2) | 242 (28.8) | 1.11* | 1.19 | 0.99 | 1.44 | 0.07 |
| Child sex | Male | 3090 (76.1) | 968 (23.9) | 1 | 1 | 1 | 1 | 1 |
| | Female | 2697 (69.9) | 1164 (30.1) | 1.38* | 1.43 | 1.29 | 1.59 | 0.001 |
| Child birth type | Single birth | 5686 (73.5) | 2055 (26.5) | 1 | 1 | 1 | 1 | 1 |
| | 1st of multiple | 56 (60.2) | 37 (39.8) | 1.83* | 2.18 | 1.41 | 3.37 | 0.001 |
| | 2nd of multiple | 45 (52.9) | 40 (47.1) | 2.46* | 2.92 | 1.86 | 4.58 | 0.001 |
| Maternal education | Unable to read & write | 3632 (69.9) | 1563 (30.1) | 2.37* | 1.46 | 1.56 | 2.17 | 0.04 |
| | Primary | 1549 (78.3) | 429 (21.7) | 1.53* | 1.06 | 0.74 | 1.62 | 0.65 |
| | Secondary | 391 (79.5) | 101 (20.5) | 1.42* | 1.26 | 0.83 | 1.92 | 0.28 |
| | Higher | 215 (84.6) | 39 (15.4) | 1 | 1 | 1 | 1 | 1 |
| Delivery by CS (Cesarean section) | No | 5653 (72.9) | 2097 (27.1) | 1 | 1 | 1 | 1 | 1 |
| | Yes | 134 (79.3) | 35 (20.7) | .70* | 1.07 | 0.71 | 1.62 | 0.73 |
| Fast, rapid breathing | No | 5279 (73.5) | 1901 (26.5) | 1 | 1 | 1 | 1 | 1 |
| | Yes | 508 (68.7) | 231 (31.3) | 1.26* | 1.22 | 1.02 | 1.45 | 0.03 |

(*Continued*)

**Table 2.** (*Continued*)

| Variable | Categories | Proximate low birth weight(n = 7919) | | OR | | 95% CI | | P-value |
|---|---|---|---|---|---|---|---|---|
| | | Average and above | Small to average | COR | AOR | Lower | Upper | |
| Delivery place | Home | 3781 (70.6) | 1576 (29.4) | 1.50* | 1.08 | 0.92 | 1.27 | 0.33 |
| | Health facility | 2006 (78.3) | 556 (21.7) | 1 | 1 | 1 | 1 | 1 |
| KMC (Kangaroo mother care) | No | 4227 (71.2) | 1707 (28.8) | 1.48* | 1.12 | 0.96 | 1.30 | 0.16 |
| | Yes | 1560 (78.6) | 425 (21.4) | 1 | 1 | 1 | 1 | 1 |
| Maternal age | 15–19 | 158 (59.6) | 107 (40.4) | 1.54* | 1.86 | 1.19 | 2.92 | 0.007 |
| | 20–24 | 1111 (70.6) | 462 (29.4) | 0.95 | 1.24 | 0.85 | 1.82 | 0.27 |
| | 25–29 | 1765 (73.7) | 630 (26.3) | 0.81 | 1.01 | 0.70 | 1.47 | 0.10 |
| | 30–34 | 1351 (74.6) | 461 (25.4) | 0.78* | 0.94 | 0.64 | 1.36 | 0.72 |
| | 35–39 | 942 (76.2) | 295 (23.8) | 0.71* | 0.83 | 0.57 | 1.22 | 0.35 |
| | 40–44 | 353 (73.1) | 130 (26.9) | 0.84 | 0.92 | 0.61 | 1.39 | 0.68 |
| | 45–49 | 107 (69.5) | 47 (30.5) | 1 | 1 | _1 | _1 | _1 |

* = p-value <0.05 for crude odds ratio.

report, although we consider it as a proximate birth weight. However, there are a number of evidences that indicated the presence of good consistency between re-called measured birth weights and mothers subjective evaluation of their children birth size. The evidence concluded that mothers' reported birth size as very small or smaller than average can be used as a reasonably good indicator of LBW [30,34–36]. The evidence might be as a result of that mother's ignorant behavior of the actual weight of the baby or might have difficulty in re-calling the actual birth weight at the data collection time of the 2016 EDHS. But, mothers can easily recall the size of the baby. With such evidences and justifications, using the mothers perceived child size as proxy indicator of birth weight could not underestimate or overestimate the prevalence of LBW (very small and small child size). Accordingly, the prevalence of proximate LBW in this study was 26.9% (95%CI), (25.4, 27.9).

The present study's prevalence of PLBW is consistent with a study conducted in Uganda 25.5% [37], and a national survey in Ethiopia indicated that 27.9% [38] of neonates were had low birth weight. This might be because of mothers' accuracy in identifying their child size as very small or small or average or large or very large irrespective of geographic and cultural variation. In addition, there might be a similarity in their socio-demographic status between Ethiopia and Uganda. But the prevalence of proxy LBW in this study is lower than the 2011 EDHS based study [39] that reported 29.1% of proxy LBW [14], and other primary studies conducted in the country, Ethiopia, that reported 28.3% [40] and 34.2% of LBW from Oromia [41], and 54.0% of LBW [42] from Tigray. The variation of the present study against those former studies might be because of study period, which might be natural cause for socio-economic improvement. Almost all the former papers studied before 2012, which have 4 years variation with the current study. The prevalence of proxy LBW in this study is higher than other study conducted in Bangladesh (20%) [43]. The current study's prevalence of PLBW also higher than studies conducted in Ethiopia that reported 21% in Dire-Dawa [44], 21.2% in Bahirdar [45], 22.5% in Jimma [46], and 21.9% [15] in eastern Ethiopia. The difference might be as a result of socio-economic distinction. Almost all the studies including the study from Afghanistan studied in major cities than the present study that considers both urban and rural areas proportionally. Maternal age, 15–19 years old (AOR), (95%CI), (1.86), (1.19, 2.9) increase the odds of proxy LBW. The finding was similar with a studies conducted in Nepal [47], Ethiopia, Bale [48], north Wollo, Ethiopia [49], and the 2011 EDHS based study [39]. A review on LBW in

Ethiopia also reported that maternal age of 15–19 years old increase the odds of low birth weight [20]. The similarity between these studies might be as a result of physiological, psychological, and behavioral immaturity of such age group mothers that affect their children in providing sufficient nutrition and care. Maternal education (1.46), (1.56, 2.17) was associated with proxy LBW. But studies conducted in Dire-diwa [44], and northwest Ethiopia did not report such association [50]. Other hospital based study also denied the association of maternal education and proxy LBW [51]. The variability might be as a result of sample size, and study setting difference. The former studies were confined to health facilities or single community.

However the 2011 EDHS based study reported an association between maternal education and proxy LBW [39]. The similarity might be as result of methodological, sample size and study population correspondence. Multiple birth during the first parity (2.18), (1.41, 3.37), and second parity (2.92), (1.86, 4.58) were associated with proxy LBW than single birth. The present study's association is contradict with a study conducted in northwest Ethiopia [50]. Although, the former study did not report the association either because of sample size or population difference, the association between PLBW and multiple birth during a first parity is anticipated. The justification is that twins shared nutrients from a single mother than singleton birth that get independently. The present study reported that female sex was associated with proxy LBW (1.43), (1.29, 1.59). The result is consistent with a study conducted in Bangladesh (43). But the association between sex and PLBW is not concordant with a study conducted in Ethiopian hospitals [51], as well as a study from Addis Ababa did not report such association [52]. However, a study from other Ethiopian hospitals showed that female births were more likely to have LBW [51]. The possible explanation for having low birth weight female children might be as a result of sex preference. Many couples prefer male, and provide adequate care while the conception is sons than daughters. Wanted latter birth (1.26), (1.09, 1.47) increase the odds of proxy LBW than the planned birth. This finding is not similar with a study that conducted in Dire-dawa in which the association was not existed for birth preparedness [45]. However, a study from Addis Ababa hospitals indicated the association of proxy LBW, and birth preparedness. The study reported that planned birth prevent the occurrence of proxy LBW by 70% than unplanned birth [52]. The possible justification for having LBW when the pregnancy is unplanned might be socio-economic, psychological, behavioral, or job related factors. Rural residence (1.39), (1.07, 1.81) increase the odds of proxy LBW than neonates born in urban settings. The association is similar with the 2011 EDHS based study that showed neonates from rural residents were 1.32 times more likely to be proxy LBW [39]. However, the present study's association is discordant with a study that conducted in Dire-dawa that did not show an association [44]. The similarity in having association regarding residence might be due to similar socio-economic factors. But the difference might be as a result of study setting difference.

In this study, region was associated with proxy LBW. Children who were born from Afar region (2.44), (1.82, 3.27) predisposed to proxy LBW than Dire-dawa children. However children from Somalia region (0.73), (0.55, 0.97), Benishangul-Gumz (0.48), (0.35, 0.67), and SNNPR region (0.67), (0.48, 0.93) were less likely to have proxy LBW. This agreed with a study that was a further analysis from the 2011 EDHS data. The report indicated that neonates born from Gambella and Somalia region were less likely to be proxy LBW [39]. The possible justification could be maternal factors. Mothers from Afar region might be stunted or had under nutritional status than Dire-dawa, and the reverse might be true for Somalia and Benishangul-Gumz. In addition, the socio-economic status of the regions might be contributed to have low birth weight children. For example, the socio-economic class of Afar, Somalia, and SNNPR is lower compared to Dire-Dawa [19]. Children who had short and rapid breathing at time of birth (1.22), (1.02, 1.45) were increasing the odds of proxy LBW. This might be as result of that

short and rapid breathing is a manifestation of intra-uterine infection or other problem that could contribute to lose child birth weight. Being a follower of protestant religion prevent proxy LBW by 24% (0.76), (0.60, 0.95). Unlike to the present study that failed to show the association of Khat chewing and proxy LBW, a study from Bale reported that Khat chewing increases the odds of proxy LBW [48]. This contradicted with a study that conducted in north Wollo and reported Khat chewing was not associated with proxy LBW [49]. In this further analysis, wealth index was not associated with proxy LBW. But the 2011 EDHS based study reported the association of LBW and wealth quintiles [39]. The difference might be the sample size. The present study used relatively a large sample size than the former study.

## Conclusion

The national proximate LBW prevalence, as indicated by very small and small child size, is high. This indicates that the national and local health facilities should mitigating measures on low birth weight. The trends of proxy LBW from 2011 to 2016 indicate almost constantly high, although there was a slight (0.5%) decrement from 2011 in relative to other study years (2012–2016). Although, the Ethiopian government and partners like WHO and Ethiopian pediatrics association appreciates the severity of low birth weight, the measures they take is not sufficient and not supported by evidence [53]. Thus, for effective measurement, the government and partners should emphasis more on investigating the causes of LBW simultaneously those organizations should develop a draft guideline to prevent LBW. Thus, this paper will be used by ministry, health facilities, and volunteer organizations to develop and amend policies to treat or prevent low birth weight in the country.

## Limitation

The limitation of this study is the subjectivity of the data that might limit the precision of the findings. Due to this, all the outcomes of this further analysis are considered as proximate results. However, this manuscript try to get a nation based evidence on proxy LBW, which will be used for programmers while designing a health education.

## Acknowledgments

### Declarations

We would like to acknowledge the central statistical agency of Ethiopia for the data (EDHS) they collected and provide it online in collaboration with the USAID.

## Author Contributions

**Conceptualization:** Mesfin Wudu Kassaw, Ayele Mamo Abebe, Ayelign Mengesha Kassie, Biruk Beletew Abate, Seteamlak Adane Masresha.

**Data curation:** Mesfin Wudu Kassaw, Ayele Mamo Abebe, Ayelign Mengesha Kassie, Biruk Beletew Abate, Seteamlak Adane Masresha.

**Formal analysis:** Mesfin Wudu Kassaw, Ayele Mamo Abebe, Ayelign Mengesha Kassie, Biruk Beletew Abate, Seteamlak Adane Masresha.

**Funding acquisition:** Mesfin Wudu Kassaw, Ayele Mamo Abebe, Ayelign Mengesha Kassie, Biruk Beletew Abate, Seteamlak Adane Masresha.

**Investigation:** Mesfin Wudu Kassaw, Ayele Mamo Abebe, Ayelign Mengesha Kassie, Biruk Beletew Abate, Seteamlak Adane Masresha.

**Methodology:** Mesfin Wudu Kassaw, Ayele Mamo Abebe, Ayelign Mengesha Kassie, Biruk Beletew Abate, Seteamlak Adane Masresha.

**Project administration:** Mesfin Wudu Kassaw, Ayele Mamo Abebe, Ayelign Mengesha Kassie, Biruk Beletew Abate, Seteamlak Adane Masresha.

**Resources:** Mesfin Wudu Kassaw, Ayele Mamo Abebe, Ayelign Mengesha Kassie, Biruk Beletew Abate, Seteamlak Adane Masresha.

**Software:** Mesfin Wudu Kassaw, Ayele Mamo Abebe, Ayelign Mengesha Kassie, Biruk Beletew Abate, Seteamlak Adane Masresha.

**Supervision:** Mesfin Wudu Kassaw, Ayele Mamo Abebe, Ayelign Mengesha Kassie, Biruk Beletew Abate, Seteamlak Adane Masresha.

**Validation:** Mesfin Wudu Kassaw, Ayele Mamo Abebe, Biruk Beletew Abate, Seteamlak Adane Masresha.

**Visualization:** Mesfin Wudu Kassaw, Ayele Mamo Abebe, Ayelign Mengesha Kassie, Biruk Beletew Abate, Seteamlak Adane Masresha.

**Writing – original draft:** Mesfin Wudu Kassaw, Ayele Mamo Abebe, Ayelign Mengesha Kassie, Biruk Beletew Abate, Seteamlak Adane Masresha.

**Writing – review & editing:** Mesfin Wudu Kassaw, Ayele Mamo Abebe, Ayelign Mengesha Kassie, Biruk Beletew Abate, Seteamlak Adane Masresha.

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
