## [Decision Letter · Decision Letter 0]

20 Nov 2020

PONE-D-20-27909

Trends of proximate low birth weight and associations among children under-five years of age : Evidence from the 2016 Ethiopian demographic and health survey data

PLOS ONE

Dear Dr. Kassaw,

Thank you for submitting your manuscript to PLOS ONE. After careful consideration, we feel that it has merit but does not fully meet PLOS ONE’s publication criteria as it currently stands. Therefore, we invite you to submit a revised version of the manuscript that addresses the points raised during the review process.

We look forward to receiving your revised manuscript.

Kind regards,

William Joe

Academic Editor

PLOS ONE

Journal Requirements:

2. Please include your tables as part of your main manuscript and remove the individual files. Please note that supplementary tables (should remain/ be uploaded) as separate "supporting information" files

3. Please list the name and version of any software package used for statistical analysis, alongside any relevant references. For more information on PLOS ONE's expectations for statistical reporting, please see https://journals.plos.org/plosone/s/submission-guidelines.#loc-statistical-reporting.

Reviewers' comments:

Reviewer's Responses to Questions

**Comments to the Author**

1. Is the manuscript technically sound, and do the data support the conclusions?

Reviewer #1: Yes

Reviewer #2: Yes

2. Has the statistical analysis been performed appropriately and rigorously? 

Reviewer #1: No

Reviewer #2: Yes

3. Have the authors made all data underlying the findings in their manuscript fully available?

Reviewer #1: Yes

Reviewer #2: Yes

4. Is the manuscript presented in an intelligible fashion and written in standard English?

Reviewer #1: No

Reviewer #2: Yes

5. Review Comments to the Author

Reviewer #1: 1)In abstract result is not presented properly instead only statistics are given. please write the major findings in sentence.

2)In method section you have listed a set of of characteristics but you have not given any justification why you refer to these characteristics, are they from your literature study, if so mention clearly. please highlight the specific research gap you have tried to address in your work.

3) The result section is presented in a casual manner and table statistics were presented only. please write in words what they imply.

4) If information on birth weight is available in the data set then can you use this variable as birth size may be subjective as it is based on perception of the mother may be or that of respondent.

5) The set of of explaining variables in the table 2 does not give justification why you choose them, instead you can run a factor analysis and find out the most important ones. some of these explaining variables might have association between them.

6) I suggest authors should use factor analysis to identify which explaining variables are related to the individual characteristics (maternal education, khat chewing, child sex, sex of the child, child birth type etc), household characteristic (wealth, residence religion etc) and policy variable (place of delivery, delivery attended by whom??).

7) a justification for using binomial logit can be provided.

8) the language in the text needs to be refined. please take help of professional to refine the text. for example what do you mean by 'child sex', instead sex of the child would be better.

9) In conclusion section there is no reference to how this research contributes to policy for child health. please include some policy implication.

10)Your region variable is very important and you have not shown how different regions in Ethopia vary in socioeconomic profile. It will help to build context for regional variation in your analysis.

11) Please give the model statistics to show the robustness of your model fit.

12) please write a section on policies adopted in Ethopia on child health and what can be done to improve them for better child health outcome in terms of birth weight and size of the child at birth.

Reviewer #2: I really appreciate the efforts by the author(s) on this manuscript but I feel the paper is not yet ready for publication in current shape and needs some major revisions. My comments are as follows:

1. The introduction should aim to contextualize the topic for the empirical analysis from the evidence with focus on similar studies from Ethiopia that were conducted in recent years.

2. The authors should present bivariate analysis to see the relationship and outcome of independent and dependent variables rather than directly attempting the regression analysis. This analysis will help understand the correlations across determinants and its association with low birth weight.

3. I also found the author(s) is missing many of the maternal health factors which are affecting the birth weight. For example, maternal BMI, anemia status, maternal nutrition and indicators such as antenatal care etc.

4. The policy implications of the findings should be discussed in the context of the literature and policy focus in Ethiopia.

5. The trend analysis should be highlighted in the introduction or in the discussion section.

Author should incorporate the comments and revise the manuscript and submit again as I think the topic has potential to be published but not in current form.

6. PLOS authors have the option to publish the peer review history of their article (what does this mean?). If published, this will include your full peer review and any attached files.

Reviewer #1: **Yes: **RUDRA NARAYAN MISHRA

Reviewer #2: No

---

## [Author Response · Author response to Decision Letter 0]

27 Nov 2020

Responses for reviewer comments

First of all, we would like to say thank you for your constructive and very important comments.

General comments

We accept the comments and take appropriate changes. We have also edited the language 

Reviewer #1: 

1. In abstract, result is not presented properly instead only statistics are given. Please write the major findings in sentence.

We write the prevalence, and the variation in child size. In addition we have also listed the associated variables. But we may not able to show the findings clearly. The major findings are presented with major statistics. 

2. In method section you have listed a set of characteristics but you have not given any justification why you refer to these characteristics, are they from your literature study, if so mention clearly. Please highlight the specific research gap you have tried to address in your work.

We select the characteristics based on literature but if the variables we select are not available at the EDHS data, we forced to omit it. We tried to add the gaps regarding this research in last paragraph of the introduction part. 

3. The result section is presented in a casual manner and table statistics were presented only. Please write in words what they imply.

We write some of the contents of the table in narration as per the comment. But we are unable to write all the contents from the table to text. 

4. If information on birth weight is available in the data set then can you use this variable as birth size may be subjective as it is based on perception of the mother may be or that of respondent.

The birth weight in the data set is not complete. There are many missing variables even mothers cannot remember the exact birth weight but they can remember the size easily. And almost many of mothers gave a date regarding the average size. But, if there is a data of birth weight, we did not use it because our operational definition does not consider it. We take the data reported from mothers or care givers as per the EDHS inclusion criteria 

5. The set of explaining variables in the table 2 does not give justification why you choose them, instead you can run a factor analysis and find out the most important ones. Some of these explaining variables might have association between them.

Sorry for missing the clarification of variable selection to multivariate analysis but we included in this version at the first paragraph, first line of factors associated with proximate LBW section. We select variables that have a p-value of 0.25 or below, and we run mlticollinearity test between close variables based on literature and authors knowledge. In this scenario, Khat chewing and cigarette smoking were had collinearity. Thus, we select Khat chewing to the multivariate analysis. However, none of the other variables have no collinearity and we tried some of the variables randomly. 

6. I suggest authors should use factor analysis to identify which explaining variables are related to the individual characteristics (maternal education, khat chewing, child sex, sex of the child, child birth type etc), household characteristic (wealth, residence, religion etc) and policy variable (place of delivery, delivery attended by whom?).

We accept the comment and we tried a multicollinearity test for the association in between them. In the entire variables, there was no association as the Tolerance (VIF) indicates except Khat chewing and cigarette smoking. 

7. A justification for using binomial logit can be provided.

The outcome variables are binary, and fulfill the assumption of logistic regression like normality test, and large sample size, and model fitness test. We included this justification in the data analysis section. 

8. The language in the text needs to be refined. Please take help of professional to refine the text. For example what do you mean by 'child sex', instead sex of the child would be better.

Accepted and get an English language professional for final edition. 

9. In conclusion section there is no reference to how this research contributes to policy for child health. Please include some policy implication.

Accepted and try to include a policy implication using data from Minister of Health, Ethiopia and other find. We have also cited as per the comment. 

10. Your region variable is very important and you have not shown how different regions in Ethiopia vary in socioeconomic profile. It will help to build context for regional variation in your analysis

Thank you, most of the regions have association with PLBW as our multivariate analysis indicates but we did not describe it as per the socio-economic perspective. However, we include a justification with citation in the discussion part.

11. Please give the model statistics to show the robustness of your model fit.

We include it in data analysis section that the model is fitted as per the Hosmer-lemeshow-goodness of fit test. 

12. Please write a section on policies adopted in Ethiopia on child health and what can be done to improve them for better child health outcome in terms of birth weight and size of the child at birth.

We have added such section in the background 

Reviewer #2: 

1. The introduction should aim to contextualize the topic for the empirical analysis from the evidence with focus on similar studies from Ethiopia that were conducted in recent years.

We accept the comment and we cited some studies that conducted in Ethiopia. 

2. The authors should present bivariate analysis to see the relationship and outcome of independent and dependent variables rather than directly attempting the regression analysis. This analysis will help understand the correlations across determinants and its association with low birth weight.

In table two, we include the cross-tab result, the binary result (COR), and the multilevel regression (AOR) result. We did not directly run multilevel analysis, but to include the details of the binary result, the table cannot be managed. However, we include the most important components even for authors who want to done meta-analysis we include the cross-tab (A, B, C & D) variables. For selecting variables for multivariate analysis based on COR, we indicate in (*). Thus, we cannot manage all the binary and multivariate results in one table rather we prefer to present the most important contents based on some other researches. 

3. I also found the author(s) is missing many of the maternal health factors which are affecting the birth weight. For example, maternal BMI, anemia status, maternal nutrition and indicators such as antenatal care etc.

I really sorry for not answering this question by adding the variables, but these variables were excluded at the beginning of the study when we refine the data. The major limitation of DHS analysis is losing such important variables. When we select the outcome variables, we forced to lose some variables simultaneously. In addition, some of variables like BMI excluded because of missing data on the variable. 

4. The policy implications of the findings should be discussed in the context of the literature and policy focus in Ethiopia.

We accepted it and include a policy indication in the conclusion section with citation 

5. The trend analysis should be highlighted in the introduction or in the discussion section.

We tried to indicate the trend as follow 

The aim of this study was to assess the trends of proximate LBW and associations from 2011 to 2016 in Ethiopia using the 2016 EDHS data. Since the survey had 5 years compressive data, it can indicate the trends better than the other primary and area specific studies.

---

## [Decision Letter · Decision Letter 1]

4 Jan 2021

PONE-D-20-27909R1

Trends of proximate low birth weight and associations among children under-five years of age: Evidence from the 2016 Ethiopian demographic and health survey data

PLOS ONE

Dear Dr. Kassaw,

Thank you for submitting your manuscript to PLOS ONE. After careful consideration, we feel that it has merit but does not fully meet PLOS ONE’s publication criteria as it currently stands. Therefore, we invite you to submit a revised version of the manuscript that addresses the points raised during the review process.

We look forward to receiving your revised manuscript.

Kind regards,

William Joe

Academic Editor

PLOS ONE

Reviewers' comments:

Reviewer's Responses to Questions

**Comments to the Author**

1. If the authors have adequately addressed your comments raised in a previous round of review and you feel that this manuscript is now acceptable for publication, you may indicate that here to bypass the “Comments to the Author” section, enter your conflict of interest statement in the “Confidential to Editor” section, and submit your "Accept" recommendation.

Reviewer #1: All comments have been addressed

2. Is the manuscript technically sound, and do the data support the conclusions?

Reviewer #1: Yes

3. Has the statistical analysis been performed appropriately and rigorously? 

Reviewer #1: Yes

4. Have the authors made all data underlying the findings in their manuscript fully available?

Reviewer #1: Yes

5. Is the manuscript presented in an intelligible fashion and written in standard English?

Reviewer #1: Yes

6. Review Comments to the Author

Reviewer #1: you can give the equation for your logistic model in the final version of the paper. Also write bit more about policy implication of your paper. In the introduction write few sentence about how this research is bridging any existing gap in the available research on the topic.

7. PLOS authors have the option to publish the peer review history of their article (what does this mean?). If published, this will include your full peer review and any attached files.

Reviewer #1: **Yes: **RUDRA NARAYAN MISHRA

---

## [Author Response · Author response to Decision Letter 1]

11 Jan 2021

Responses for reviewer comments

First of all, we would like to say thank you for your constructive and very important comments.

I included the following sentences as per the reviewer comments 

You can give the equation for your logistic model in the final version of the paper. 

1. 

In the introduction write few sentences about how this research is bridging any existing gap in the available research on the topic. 

2. Because of that most previous conducted studies are limited geographically, by population, and sample size. In addition, those studies are single shot and did not show the trends of low birth weight which might be susceptible to bias because of drought or other seasonal factors like political instability and natural disaster. However, this study considered a 5 consecutive year’s data to assess the trends and potential determinants of proximate low birth weight. 

Also write bit more about policy implication of your paper

3. Thus, this paper will be used by ministry, health facilities, and volunteer organizations to develop and amend policies to treat or prevent low birth weight in the country. 

Thank you!!

---

## [Editor Report · Decision Letter 2]

22 Jan 2021

Trends of proximate low birth weight and associations among children under-five years of age: Evidence from the 2016 Ethiopian demographic and health survey data

PONE-D-20-27909R2

Dear Dr. Kassaw,

We’re pleased to inform you that your manuscript has been judged scientifically suitable for publication and will be formally accepted for publication once it meets all outstanding technical requirements.

Kind regards,

William Joe

Academic Editor

PLOS ONE
---

## [Editor Report · Acceptance letter]

29 Jan 2021

PONE-D-20-27909R2 

Trends of proximate low birth weight and associations among children under-five years of age: Evidence from the 2016 Ethiopian demographic and health survey data 

Dear Dr. Kassaw:

I'm pleased to inform you that your manuscript has been deemed suitable for publication in PLOS ONE. Congratulations! Your manuscript is now with our production department. 

Kind regards, 

on behalf of

Dr. William Joe 

Academic Editor

PLOS ONE